# Noncontact and High-Precision Sensing System for Piano Keys Identified Fingerprints of Virtuosity

**DOI:** 10.3390/s22134891

**Published:** 2022-06-29

**Authors:** Takanori Oku, Shinichi Furuya

**Affiliations:** 1Sony Computer Science Laboratories Inc., 3-14-13 Higashigotanda, Shinagawa-ku, Tokyo 1410022, Japan; furuya@csl.sony.co.jp; 2NeuroPiano Institute, 13-1 Hontorocho, Shimogyo Ward, Kyoto 6008086, Japan; 3Yotsuya Campus, Sophia University, 7-1 Kioicho, Chiyoda-ku, Tokyo 1028554, Japan

**Keywords:** piano, sensing system, motor skill, virtuosity

## Abstract

Dexterous tool use is typically characterized by fast and precise motions performed by multiple fingers. One representative task is piano playing, which involves fast performance of a sequence of complex motions with high spatiotemporal precision. However, for several decades, a lack of contactless sensing technologies that are capable of precision measurement of piano key motions has been a bottleneck for unveiling how such an outstanding skill is cultivated. Here, we developed a novel sensing system that can record the vertical position of all piano keys with a time resolution of 1 ms and a spatial resolution of 0.01 mm in a noncontact manner. Using this system, we recorded the piano key motions while 49 pianists played a complex sequence of tones that required both individuated and coordinated finger movements to be performed as fast and accurately as possible. Penalized regression using various feature variables of the key motions identified distinct characteristics of the key-depressing and key-releasing motions in relation to the speed and accuracy of the performance. For the maximum rate of the keystrokes, individual differences across the pianists were associated with the peak key descending velocity, the key depression duration, and key-lift timing. For the timing error of the keystrokes, the interindividual differences were associated with the peak ascending velocity of the key and the inter-strike variability of both the peak key descending velocity and the key depression duration. These results highlight the importance of dexterous control of the vertical motions of the keys for fast and accurate piano performance.

## 1. Introduction

One of the most representative features of skillful motor actions, such as surgery and musical performance, is dexterous tool use at high speed with high precision. This activity is challenging, particularly for individuals without any history of extensive manual training, due to the trade-off between speed and precision of movements [1] and thus requires people to undergo years of training to overcome it. A precise description of such skillful behaviors is essential for elucidating biomechanical principles governing the production of movements and neuroplastic mechanisms subserving the acquisition and loss of skills through training and the development of disorders [2]. A methodological challenge for such a precise description is difficulty in obtaining accurate measurements of fast and subtle movements in dexterous tool use, in contrast to gross and slow movements used in daily activities, such as grasping. Modern technologies for sensing human motions, such as motion capture with multiple high-speed cameras [3,4,5] and data gloves with multiple bending sensors [6,7,8], have enabled quantitative assessment of complex manual movements. However, the time resolution of these sensors is generally not enough to capture complex patterns of fast motions of a tool that can be manipulated in skillful motor actions, such as throwing a baseball and playing musical instruments. In addition, an occlusion of markers attached to the body and the tool has been a bottleneck to obtaining precise measurements of complex movements involving dynamic postural changes with motions at multiple joints [9]. The development of novel sensing technologies has attempted to solve such problems. For example, a miniature magnetic sensor successfully recorded motions of a ball on the order of milliseconds, and a series of experiments with it uncovered various features of skillful ball-throwing motions [10,11]. However, sensors attached to a tool can alter physical properties, such as weight and inertia, and affect tactile and proprioceptive feedback in motion, the latter of which matters particularly in the assessment of the symptoms of focal hand dystonia due to sensory trick [12,13]. Therefore, the development of contactless or noncontact sensors that enable one to record motions of a tool to be manipulated at high spatiotemporal resolution is needed to fully unveil expert motor skills in dexterous tool use.

Piano playing can be one of the most representative dexterous skills to perform [14,15,16,17]. Previous studies developed some sensors, such as pressure sensors that were placed on the bottoms of the keys [18], force sensors that were implemented on the key surfaces [19,20,21,22], and custom-made data gloves [6,23,24], which successfully recorded the motions and/or force of the piano keys and fingers at high spatiotemporal resolution. However, none of these original sensors were no-contact sensors that enabled the recording of key motions without altering the touch sensation. In contrast, the noncontact sensing technology that has been used most frequently in previous studies was the Musical Instrument Digital Interface (i.e., MIDI) [25,26,27,28,29,30,31,32,33]. This technology captures timing and seven-bit quantized key speed at only two discrete events of the key motion, the moments when the key is depressed and released [34], which provides no high spatiotemporal resolution and, therefore, fails to capture various features of fine motor control.

Here, we propose a novel sensing system capable of capturing the time course of the vertical position of 88 piano keys, without any physical contact with the keys, with a time resolution of 1 ms and a spatial resolution of 0.01 mm. The sensors were embedded under the piano keys and did not have any mechanical contact with any keys. To test whether this sensing system allows for the identification of the motor proficiency of expert pianists, a behavioral experiment with expert pianists was performed, and a set of motor engrams of each pianist’s touches was extracted from the collected data and analyzed by a penalized regression model. The results identified a novel motor skill that explains individual differences in both the maximum speed and timing precision across pianists’ fast piano performances.

## 2. Materials and Methods

### 2.1. Participants

Forty-nine expert right-handed classical pianists (41 females, 20–45 years old) without a history of serious physical problems related to piano playing served as the participants in the present study. Most of the participants were pianists who studied at music conservatories in Japan. Each pianist underwent at least 15 years of piano training at music conservatories and/or privately under the supervision of professional pianists. In accordance with the Declaration of Helsinki, the experimental procedures were explained to all participants. Informed consent was obtained from each participant prior to the experiment. The Ethics Committee at Sophia University approved this study.

### 2.2. Sensing System

The vertical position of each key was measured using a custom-made contactless optical sensor system (Figure 1A). The sensor system was mounted beneath the piano keys of an acoustic piano (i.e., key-bed). The sensor consists of 88 photo reflectors (LBR-127HLD, Letex Technology Corp. Taichung City, Taiwan), seven 12-bit analog-to-digital (A/D) converter-integrated circuit chips (ADS7953, Texas Instruments, Dallas, TX, USA), and a microprocessor (STM32F446, STMicroelectronics, Geneva, Switzerland). Each of the photo-reflective sensors beneath the key projects infrared light on the bottom surface of the key, and the derived voltage signal changes in relation to the intensity of the reflected infrared light. The intensity of the reflected infrared light increases linearly with a decrease in the distance between a photo reflector and the bottom surface of a key (see details in the Results). Therefore, the change in the voltage signal represents the change in the distance between a photo reflector and the bottom surface of a key (Figure 1B). Each A/D converter controlled by the microprocessor collects the voltage signal from 12 or 16 photo reflectors. The voltage signal is stored on a personal computer via the microprocessor connected by USB Full Speed at a sampling frequency of 1 kHz and converted to the vertical distance (Figure 1C). The sensor system calculates the vertical position of each key by a linear interpolation of the sensor value between the neutral position and bottom position stored in the calibration procedure performed prior to the experiment. The sensor system can, therefore, record the time-varying vertical position of all 88 piano keys at a 0.01 mm spatial resolution without any physical contact that can affect the mechanical characteristics of the piano keystroke.

### 2.3. Experimental Setup and Task

The experimental apparatus consisted of a Yamaha acoustic piano (U1) and the sensing system. Figure 2 shows the experimental task requiring the repetition of two sets of simultaneous keystrokes of the two keys, leaving one white key in between (i.e., a major third interval), using the right index and ring fingers for one set and the right middle and little fingers for another set. Participants were asked to perform the task for 6 s as fast and accurately as possible and at a paced tempo (100 bpm) with a predetermined loudness (i.e., mezzo forte), which was provided as a sound stimulus from the speaker located in front of the participant. We used this task because such a chord-trill task, which has been included in various musical pieces (e.g., Etude Op.25 no.6 by Frederik Chopin, Ondine by Maurice Ravel, Piano Sonata No.3 1st mov. by Ludwig van Beethoven), is known to be technically challenging to play quickly and accurately.

### 2.4. Data Analysis and Statistics

The position data for the keys were low-pass filtered using a second-order Butterworth filter with a cutoff frequency of 20 Hz. To identify the keystroke skill that explains the inter-individual variability of piano expertise, we performed a penalized regression analysis using spatiotemporal features of the vertical motion of the keys as independent variables and the maximum speed and loudness accuracy as dependent variables. Figure 3 illustrates six features characterizing the time-varying waveform of each vertical motion of the keys and their derivatives (i.e., velocity). Each of the features was selected to characterize different events of one cycle of the keystroke motion. The features “peak_des_vel” and “peak_to_bottom” characterize the key-descending phase, “max_depth” and “depression_ratio” characterize the key-pressing phase, and the “release_to_peak” and “peak_asc_vel” characterize the key-ascending phase, respectively. For each feature variable, the mean and standard deviation across strikes within the strikes during each performance were computed.

We used the inter-keystroke interval at the fastest tempo (a difference in timing between two successive keypresses) and loudness balance at the paced tempo (a difference in the peak descending velocity of the two keys to be depressed simultaneously) as variables representing the maximum tempo and precision of the performance, respectively. Using each of the two variables as a dependent variable and the inter-strike mean and standard deviation values of the aforementioned six features characterizing the waveforms of the vertical positions of the keys as independent variables, a penalized regression (i.e., elastic net regression) was performed to identify motor skills associated with speed and accuracy of the finger movements of expert pianists. Elastic net regression selects variables while optimizing the balance of sparsity and explainability of the model [35]. Here, each variable used for the regression was standardized (subtracting the mean value and dividing by the standard deviation). All of the analyses were performed using the library “Scikit-learn” in Python [36].

## 3. Results

### 3.1. Performance of the Sensing System

To evaluate the ground noise of the system, we computed the ratio of the maximum amplitude of the ground noise (i.e., the difference between the maximum and minimum sensor values when the key was stable at the highest position for 20 s) relative to the range of sensor values (i.e., the difference between the sensor value when the key was located at its bottom position and the sensor value when the key was at its highest position). The ratio was 0.10 ± 0.032% across all sensors. This ratio almost corresponded to the spatial resolution of the sensor system and was negligible.

To evaluate the linearity of the system, we pushed the keys down from 0 mm to 10 mm at 1 mm intervals by means of a micrometer and calculated a linear error by comparing the measured sensor values with sensor values estimated from a least squared linear regression of the measured sensor values. The across-key average of the ratio of the average linear error of each point relative to the measurement range of the sensor value was 2.4 ± 0.67%.

To evaluate the temporal precision of the system, we executed a continuous 3 min measurement of the key position 10 times and evaluated the variation and accuracy of the sampling interval time. We evaluated the variation in the sampling interval time by computing the standard deviation of the difference of sampling time recorded from the system’s internal clock. We also evaluated the deviation of sampling time between two adjacent keys by the system’s internal clock. We evaluated the accuracy of the sampling interval by comparing the difference in receipt time between the start command and the end command recorded by the system’s internal clock with the difference in the sent time between the start command and the end command recorded by the internal timer of the Windows operating system called “QueryPerformanceCounter”, which has a 1-microsecond time resolution. The average standard deviation of the sampling time was 8.08 ± 0.02 × 10^−3^ milliseconds. This deviation was less than 1% of the sampling time interval, which was negligibly small. The average of the difference in sampling time between two adjacent keys was 5.96 ± 0.04 × 10^−3^ ms. In case of the third chord used in the experiment, the difference in sampling time between top and bottom notes was about 24 × 10^−3^ ms. This was less than 2.5% of the sampling time interval, which was negligibly small. The error between the elapsed time recorded by the system’s internal clock and that recorded by the OS’s internal clock was 4.16 ± 0.50 × 10^−5^%. This error corresponded to less than half of a sampling interval time for a 10 s recording, and it was negligible enough for a short recording, such as that used in the experiment in this study.

### 3.2. Results of the Regression Model Based on the Human Experiments

Table 1 summarizes the results of the elastic net regression explaining the interindividual variance for both the maximum tempo and loudness balance across the participants by displaying the mean and standard deviation of the six features of the key movements across strikes. For the maximum tempo and the loudness balance, the *R*^2^ value derived from the model prediction based on the observed values of the feature variables was 0.852 and 0.759, respectively, which are visually displayed in Figure 4A,B. The alpha value was 0.030 and 0.032 for the model explaining the fastest tempo and loudness balance, respectively, which indicates that the model was almost the same as the ridge regression. Between the fastest tempo and loudness balance, there was no correlation (r = −0.183, *p* = 0.214), which indicated that these performance variables were independent.

Figure 4 illustrates a schematic drawing of the elastic net model explaining both the fastest speed and loudness balance according to the feature variables of the key movements of the pianists. For the model of the fastest speed (i.e., inter-onset interval: IOI), the coefficient value was large, particularly for the inter-strike mean of the peak descending velocity of the key, the inter-strike mean of the key depression ratio, and the inter-strike mean of the time to key release. For the model of loudness balance, the coefficient value was large, particularly for the inter-strike variability of the peak key descending velocity, the inter-strike variability of the key depression ratio, and the inter-strike mean of the peak ascending velocity of the key.

## 4. Discussion

In the present study, we developed a novel sensing system capable of measuring the time-varying vertical position of all piano keys with high spatiotemporal resolution without having any physical contact with the keys. By using the system, we assessed a variety of features of movements of multiple keys while expert pianists were performing fast and accurate keystrokes. By analyzing these features for a large number of pianists with a machine learning technique using a penalized regression model, we identified novel spatiotemporal features of key motion that were associated with the speed and accuracy of dexterous finger movements in piano performance. There are three major results in this study. First, we confirmed that our sensing system that has no mechanical contact with the piano keys could record the vertical position of the piano keys with 1 ms of temporal resolution and 0.01 mm of spatial resolution with a linearity between the position and voltage of the signal. Second, the system identified novel features of the key motions describing the pianists’ expertise (i.e., speed and accuracy), which were not only the timing and velocity of the key depression and release that had been assessed in many previous studies using the MIDI technology but also the spatiotemporal features of the peak velocities of key movements, the maximum displacement of key descent, and the duration during which keys were maximally depressed. Third, using a large dataset of key movements collected from 49 pianists, a penalized regression model identified a novel set of task-relevant features of the finger touches that explain the individual differences in the speed and accuracy in dexterous keystroke tasks. For the mean inter-keystroke interval representing the agility of the repetitive keystrokes, the individual differences across the pianists were associated with the key depression duration, the time to the moment when the key ascending velocity reached its peak, and the peak descending velocity of the key. For the loudness balance of the two simultaneously depressed keys representing the precision of the key-depression velocity, the individual differences were associated with the inter-strike variability of the peak descending velocity and the key-depression duration and the inter-strike average of the peak ascending velocity of the keys. These results indicate that motor skills necessary for dexterously depressing and releasing the keys play a crucial role in both fast and accurate performance of sequential finger movements in piano performance.

A close inspection of the results of the regression model deepened the understanding of mechanisms underlying fast and accurate performance of the finger motions. First, the negative covariation between the loudness balance error of the two simultaneous keystrokes and the inter-strike variability of the key depression duration suggests that pianists who changed motions in a strike-by-strike manner were better at keystroke feedback control based on afferent sensory information. In the accurate performance of repetitive piano keystrokes, previous studies have demonstrated crucial roles of feedback control based on somatosensory information [27,37,38]. In feedback control, afferent sensory information is integrated into motor commands to correct movement error. The key-depression duration indicates the duration during which the fingertip receives the reaction force originating from the mechanical interaction between the key and key-bed, whereas this duration is not at all associated with the tone loudness. It is, therefore, possible that the inter-strike variability of key depression represents a process of online correction of movements based on the somatosensory feedback derived from fingertips during repetitive keystrokes. Second, it is reasonable that the loudness balance error covaried positively with the inters-trike variability of the peak key descending velocity because the key descending velocity determines the tone loudness [20]. One possible account for this outcome is that pianists who can produce the target key-striking velocity consistently are able to discriminate subtle deviations in tone loudness or force applied to the key and, therefore, can perceive a subtle loudness error between the two tones in the key presses [37]. It is also plausible that pianists with a smaller amount of signal-dependent noise in the motor commands [39] displayed both reduced inter-strike variability of the key-striking velocity and lower loudness balance error. Third, a positive covariation between the peak ascending velocity of the key and the loudness balance error indicates that faster finger-lift motions allowed for earlier preparation of subsequent keystrokes, which may enable precise control of tone loudness. For example, preparatory auditory imagery in sequential motor actions modulates action planning to produce upcoming movements efficiently [40]. A possible implication for music pedagogy is, therefore, to encourage pianists to lift their fingers quickly following key depression for accurate loudness control of multiple tones with their fingers; this approach needs further evaluation of causality through interventional studies.

For the maximum tempo when playing at the fastest tempo, the individual differences were negatively associated with both the key depression duration and duration until the key ascending motion reached its peak velocity. These results corroborate our previous observation that the duration of the hand muscular activities during the individual keypress was negatively associated with the maximum tempo when pianists were playing as fast as possible [16]. The shortened key-depression duration and hand muscular activation, as well as quicker initiation of the key ascending movement, can allow for quicker transition of the direction of the finger movement from flexion to extension; these features are prolonged abnormally in pianists with focal hand dystonia [23,41], and thereby, skillful finger movements are impaired.

One pedagogical implication for practicing and teaching the piano can be to encourage pianists to lift their fingers quickly immediately after a key reaches its bottom to accomplish both speed and accuracy in virtuosic piano performance. When pianists are challenged to play faster or more accurately, there are many candidate skills to be taken into consideration. These include spatiotemporal features of movements of the keys and their attributes, such as movements, posture, and muscular activities of the fingers, arm, and trunk [42]. Identifying a small set of skills relevant to skillful piano performance is, therefore, useful to optimize piano practice and teaching efficiently. In the present study, a combination of the novel sensing system and machine learning analyses successfully identified only a few features of key movements, most of which were intuitively irrelevant to task performance. In future studies, it will be essential to address the causal relationship of these features through an interventional experiment to further identify crucial motor skills necessary for optimizing the musical performance of expert pianists.

From a technological point of view, the advantage of the present sensing system over various existing technologies is that it can be retrofitted into existing acoustic pianos. We have retrofitted the system into several grand pianos made by different companies (e.g., Kawai, Steinway, Yamaha), which took half an hour to complete, including the calibration process. To the best of our knowledge, the existing high-resolution sensing system should be built-in prior to shipment, which contrasts with our system. In addition, the spatiotemporal resolution of our sensing system is as high as that of other built-in sensing systems, such as Disklavier and SPIRIO.

In future works, we will perform further analyses of data with different tasks, spatiotemporal features of keystrokes, and subject groups. Identifying the spatiotemporal features relevant to the skillful performance of other basic tasks that are included in various piano pieces, such as scales and arpeggios, can help develop effective piano training. The other spatiotemporal features of keystrokes measured by our sensing system, such as the velocity at the specific moments of the keystroke, acceleration, and jerk, also may explain the skillful piano performance. For example, the velocity at the onset of the movement and collision to the key-bed may be relevant to the finger–key contact noise and key-bottom contact noise affecting the piano tone’s timbre [43]. In this study, we only evaluated healthy young pianists; however, evaluating other subject groups, such as elderly pianists and pianists with movement disorders, may help prevent the deterioration of piano performance skills with aging and movement disorders in the future study.

## Figures and Tables

**Figure 1 sensors-22-04891-f001:**
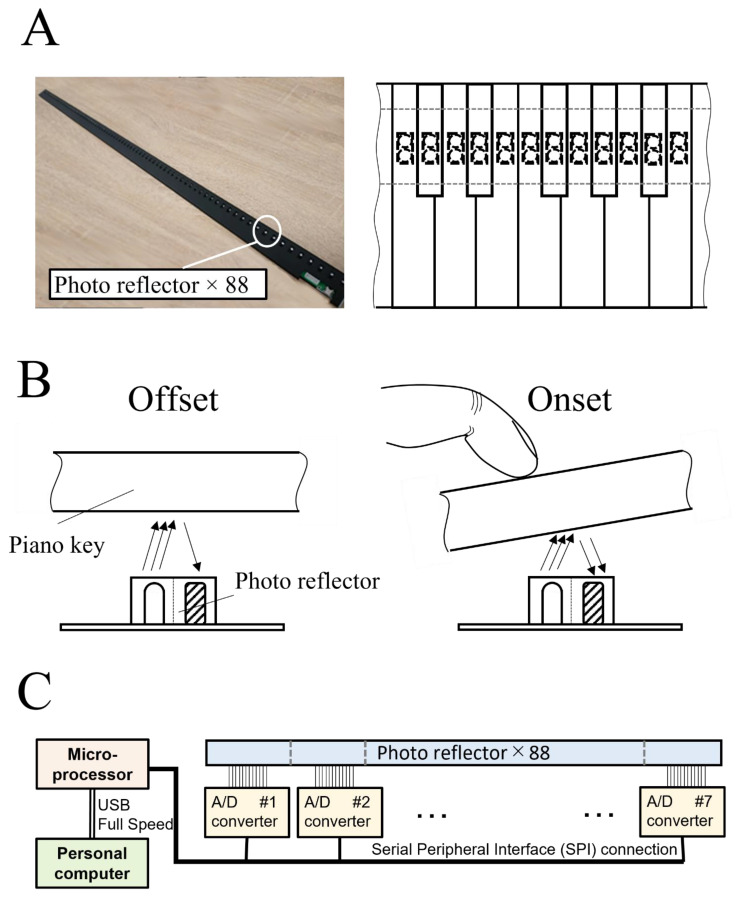
Schematic illustration of the system for sensing the key positions. (**A**) Each of the 88 photo reflectors was mounted beneath one piano key. (**B**) The photo reflector projected infrared light on the bottom surface of the key, and the derived voltage signal changed linearly with the distance between the photo reflector and the bottom surface of the key in relation to the intensity of the reflected infrared light. (**C**) System schematic of the sensing system.

**Figure 2 sensors-22-04891-f002:**
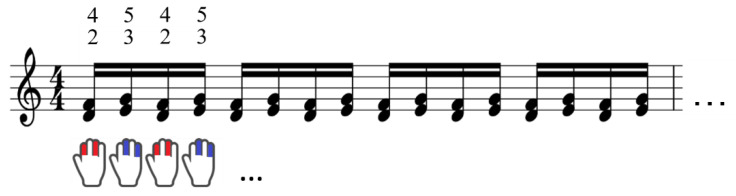
The experimental task. The experimental task required the repetition of two sets of simultaneous keystrokes of the two keys, leaving one white key in between (i.e., a major third interval), using the right index and ring fingers for one set and the right middle and little fingers for the other set.

**Figure 3 sensors-22-04891-f003:**
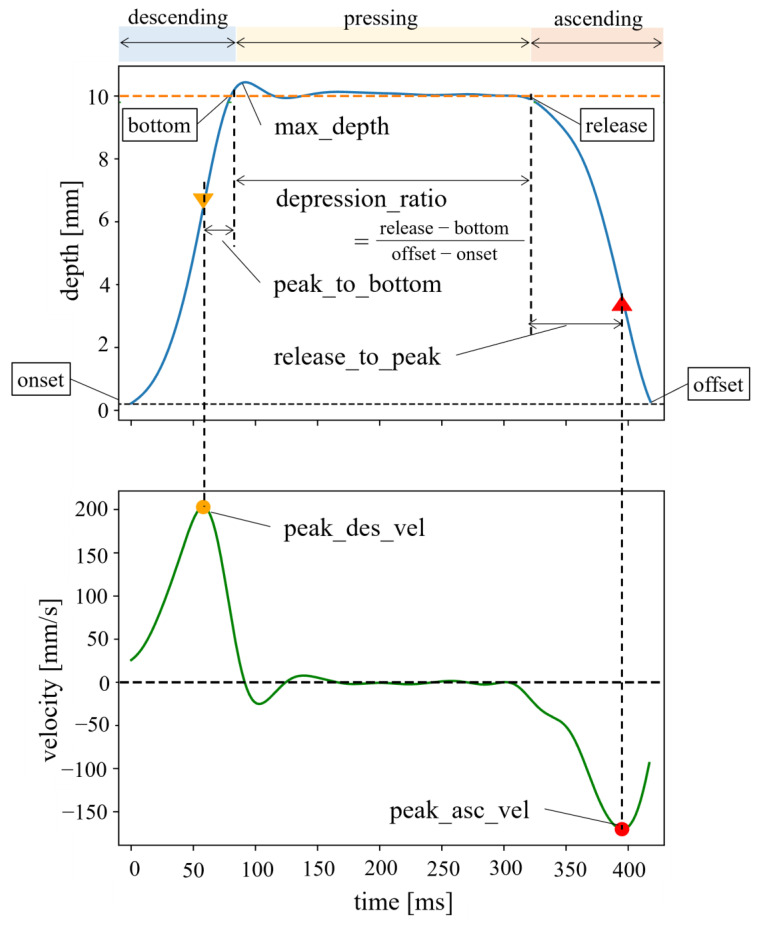
Definitions of the six characteristics of the key-depressing and key-releasing motions. (i) peak_des_vel: peak descending velocity during key depression, (ii) peak_asc_vel: peak ascending velocity during key release, (iii) max_depth: the maximum key-moving distance during key depression, (iv) peak_to_bottom: the time difference between the moment when the key reaches its peak descending velocity and the moment when the key reaches its bottom during key depression, (v) release_to_peak: the time difference between the moment when the key leaves from its bottom and the moment when the key reaches its peak ascending velocity during key lift, (vi) depression_ratio: the ratio of the duration when the key touched the bottom relative to the duration between the onset and offset of key motion.

**Figure 4 sensors-22-04891-f004:**
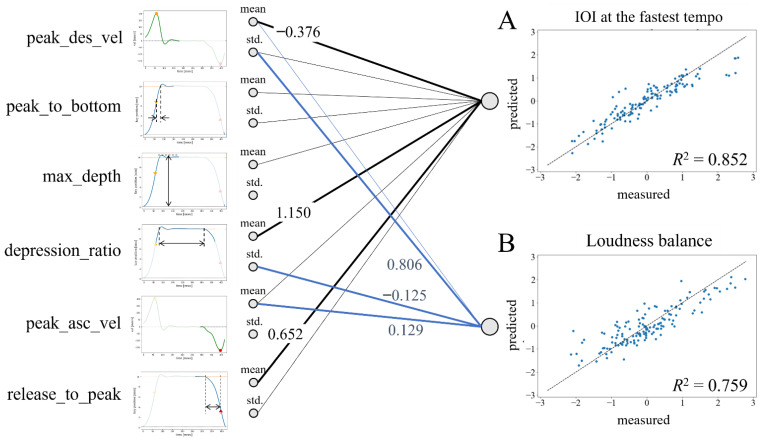
Schematics of the results of the penalized regression explaining the interindividual variance of each of the IOIs at the fastest tempo and loudness balance across the participants. (**A**) *R*^2^ of the regression model for the IOI at the fastest tempo was 0.852. The individual differences across the pianists of the IOI at the fastest tempo were mostly accounted for by the inter-strike mean of the peak descending velocity of the key, the inter-strike mean of the key-depression ratio, and the inter-strike mean of the time to key release. (**B**) *R*^2^ of the regression model for the loudness balance at the paced tempo was 0.759. The individual differences across the pianists of loudness balance was mostly accounted for by the inter-strike variability of the peak key descending velocity, the inter-strike variability of the key depression ratio, and the inter-strike mean of the peak ascending velocity of the key.

**Table 1 sensors-22-04891-t001:** Results of the elastic net regression.

	Peak Des Vel		Peak Asc Vel		Max Depth		Peak to Bottom		Release to Peak		Depression Ratio				
	Mean	Std	Mean	Std	Mean	Std	Mean	Std	Mean	Std	Mean	Std	*α*	L1 Ratio	*R* ^2^
Fastest IOI	−0.376	0.014	0.018	0.000	−0.10	0.000	0.077	0.017	0.652	0.011	1.150	0.000	0.030	1.000	0.852
Loudness Balance	0.086	0.806	0.129	0.024	0.000	0.000	0.000	0.000	0.000	0.000	0.000	−0.125	0.032	1.000	0.759

## Data Availability

The data that support the findings of this study are available from the corresponding author upon reasonable request.

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
