# Peer review of "Noncontact and High-Precision Sensing System for Piano Keys Identified Fingerprints of Virtuosity"

_sensors, 2022, doi:10.3390/s22134891_

Round 1

Reviewer 1 Report

The subject of the article is the use of a special piano key position measuring system to assess the skills of pianists. Attention was devoted to several issues: the proposed measuring system, the proposed measured dynamic parameters of the key movement and the analysis of the dependence of these parameters on the skills of pianists.
An optical measuring system was proposed here, allowing to measure the position of all 88 keys of a piano with a spatial resolution of 0.01 mm and a temporal resolution of 1 millisecond. The measuring system described by the authors consists of 88 pairs of infrared LED - phototransistor, installed under the keyboard. The LEDs illuminate the individual keys, and the intensity of the reflected light is measured by a phototransistor in a system with a 12-bit A / C converter. Measurements are made 1000 times per second and recorded by a computer. The authors report that the intensity of the reflected infrared light increases linearly with a decrease in the distance between the photoreflector and the lower surface of the key. However, this dependence is approximate due to the nonlinearities of both the elements and the measurement system itself. Calibration measurements showed an error of 2.4%, acceptable for the authors. The description of the measuring system shows that the measurements of the positions of the following keys are performed sequentially. This can introduce a slight error in the time domain, i.e. the measurements of the individual keys come from different time moments. Then the simultaneous pressing of the two keys would be registered as non-simultaneous. The authors provided the results of the analysis of the uniformity of sampling moments. It is not clear, however, what are the deviations of the measurement moments from those assumed for individual keys.

The authors used a measuring system to investigate the playing style of 49 experienced pianists. It is a large trial group. Pianists were asked to play a special music pattern. The movements of all the keys were registered, with the resolution specified earlier. On the basis of the recorded data, the authors determined six features resulting from the description of the key movement: maximum deviation from the neutral position, two velocities, three time features. Then, they analyzed the relationship of these features with the parameters that can be considered as an evaluation of the game quality: interonset interval and loudness balance (how are they measured?). It turned out that such a relationship exists and it is possible to indicate the features that show the greatest relationship with these parameters. The authors' conclusions concern, inter alia, tips for teachers, for example, encouraging pianists to quickly raise their fingers as soon as a key reaches its lowest position.

The article clearly presents the issues. In my opinion, it would be advisable to supplement the text with some information:
- how is loudness balance measured?
- the information about pressed and released keys in the MIDI standard also includes a seven-bit value specifying the speed of the key, not just the time of pressing and releasing, as stated in the article;
- have there been studies of the relationship between IOI parameters and loudness balance with the generally understood virtuosity of pianists? Ie, will we obtain a pianist's skill assessment consistent with that made by a human by examining the IOI and loudness balance?
- have the authors considered joining the feature of maximum acceleration?

A few typing errors noticed:
Experimental setup and tast -> task
we recoded the piano key motions -> recorded
Each of the photo reflective sensor -> sensors
can affect the mechanics characteristics -> mechanical
Windows operation system -> operating
Each of the features were selected -> was selected

Reviewer 2 Report

Dear authors,

First at all, let me congratulate you for the paper. There are a few things that they should be address before its publication:

- Forty-nine expert right-handed classical pianists (41 females, 20 - 45 years old) participated in the study, however it is not sufficient. Are all the participants from the same country? Why to restrict the study to pianists under 45 years old? Just eight males? It is recommended to incorporate more participants in future researchs.

- I suppose that in line 93 the authors are refering to Fig. 1.A. 

- It is highly recommended to include "future works".

- In line 209 R2 should be R^2.

- 19 references are self-references. More than the half of the total number of references.

KInd regards.
